# Auger electron emission initiated by the creation of valence-band holes in graphene by positron annihilation

V.A. Chirayath[1], V. Callewaert[2], A.J. Fairchild[1], M.D. Chrysler[1], R.W. Gladen[1], A.D. Mcdonald[1], S.K. Imam[1], K. Shastry[1,3], A.R. Koymen[1], R. Saniz[2], B. Barbiellini[4], K. Rajeshwar[5], B. Partoens[2] & A.H. Weiss[1]

Auger processes involving the filling of holes in the valence band are thought to make important contributions to the low-energy photoelectron and secondary electron spectrum from many solids. However, measurements of the energy spectrum and the efficiency with which electrons are emitted in this process remain elusive due to a large unrelated background resulting from primary beam-induced secondary electrons. Here, we report the direct measurement of the energy spectra of electrons emitted from single layer graphene as a result of the decay of deep holes in the valence band. These measurements were made possible by eliminating competing backgrounds by employing low-energy positrons (<1.25 eV) to create valence-band holes by annihilation. Our experimental results, supported by theoretical calculations, indicate that between 80 and 100% of the deep valence-band holes in graphene are filled via an Auger transition.

[1] Department of Physics, University of Texas at Arlington, Arlington, Texas 76019, USA. [2] Department of Physics, Universiteit Antwerpen, Antwerpen 2020, Belgium. [3] Department of Physics, R.V. College of Engineering, Bangalore 560059, India. [4] Department of Physics, Northeastern University, Boston, Massachusetts 02115, USA. [5] Department of Chemistry and Biochemistry, University of Texas at Arlington, Arlington, Texas 76019, USA. Correspondence and requests for materials should be addressed to V.A.C. (email: chirayat@uta.edu).

In an Auger process, the energy released when an electron from a higher energy level falls into a deeper level results in the emission of an energetic electron whose energy is characteristic of the electron levels involved. In the case of holes in core levels, the branching ratio for such a process is close to unity[1] and can result in the emission of electrons that can escape from the surface with no loss of energy. Auger electron spectroscopies based on the measurement of electrons emitted as a result of the decay of core holes produced by energetic beams of X-rays or electrons have found wide application in the analysis of surfaces[2], and direct comparisons between theoretical line shapes and experimental spectra without background subtraction have been successful for core hole Auger spectra[3,4]. However, there have been only indirect investigations of the spectra of Auger electrons emitted as a result of the filling of valence holes in solids[5,6]. A similar process involving the filling of valence levels in oxygen have been recently invoked to explain low-energy electron emission following ionization of water dimers[7,8]. The Auger spectra due to the filling of valence holes carry important information on electron correlation, surface projected density of states and the decay channels of the valence holes. The central obstacle for the direct measurement of the Auger electron spectrum produced by the valence hole decay has been the large secondary background in the low-energy regime, produced by the energetic electron or photon beams used to create the initial hole. Recent measurements in our laboratory[9] have shown that it is possible to obtain Auger spectra that are completely free of such secondary electron background by utilizing a positron beam with an incident kinetic energy below $\sim 2\,\text{eV}$ (ref. 10), the threshold for impact induced secondary electron emission.

In this study, we have used a beam of 1.25 eV positrons (antimatter electrons) to measure the spectrum of electrons emitted as a result of the filling of holes in the valence band of single-layer graphene (SLG) created by the process of matter–antimatter annihilation[11]. Using this method, we were able to measure the spectra of electrons emitted solely as a result of Auger transitions and free of beam impact-induced background down to 0 eV. Comparison of the ratio of the intensities of the Auger peaks due to the filling of valence and core holes to the theoretically calculated ratio indicates that the Auger process is a dominant channel for the decay of deep valence holes in graphene.

## Results

**Positron annihilation-induced Auger spectrum from SLG and Cu.** The time of flight (TOF)-positron annihilation-induced Auger electron spectra (PAES) of SLG deposited on a polycrystalline copper substrate is shown in Fig. 1. The TOF-PAES spectra obtained from polycrystalline Cu after the removal of single layer graphene through sputtering is also shown. Measurements revealed the existence of a strong peak at low energies in the TOF electron spectra from SLG. This peak, which occurs at $\sim 4\,\text{eV}$, is due to Auger emissions resulting from the filling of annihilation-induced deep holes in the 20 eV wide valence band of the graphene layer. The low-energy peak was notably absent in the spectrum taken from the clean Cu substrate, which is consistent with the fact that the Auger electrons resulting from the filling of holes in the relatively shallow valence band of Cu do not have enough energy to leave the surface.

The mechanism that produces this low-energy peak has been described schematically in Fig. 2. The primary hole in the valence band of graphene is created by the annihilation of a surface bound positron with electrons in the valence band (shown on the left of Fig. 2). The annihilation-induced hole is filled by a second electron in the band and results in the excitation of another electron to an unoccupied state (shown in the middle of Fig. 2). Following the nomenclature of Auger spectroscopy, we use the term VVV to refer to an Auger transition in which a hole in the valence band is filled by an electron higher-up in the valence band, resulting in the ejection of a third electron from the valence band. Thus, a VVV Auger process starts with one hole in the valence band and ends with two valence-band holes in the final state.

Neglecting correlation effects, the kinetic energy, $KE_{Auger}$, of electrons emitted into the vacuum via a VVV Auger transition can be determined from energy conservation to be:

$$KE_{Auger} = E_h - E_1 - E_2 - \phi \qquad (1)$$

where $E_h$ is the energy level of the initial hole, $E_1$ and $E_2$ are the binding energies (with respect to the Fermi level) of the electronic states involved in the transition and $\phi$ is the work function. From equation (1) it can be seen that the maximum kinetic energy for such an electron emitted into the vacuum is:

$KE_{Auger\text{-}max} = \Delta_{VB} - \phi$ , where $\Delta_{VB}$ is the valence band width (that is, the energy difference between the top and bottom of the filled states in the valence band). Equation (1) implies that it is energetically possible for an electron to be emitted into the vacuum via a VVV Auger transition from any material with $\Delta_{VB} > \phi$. In the case of graphene, the width of the valence band is $\sim 20\,\text{eV}$ (shown on the right in Fig. 2) and the value of the work function is $\sim 4.5\,\text{eV}$ (ref. 12). Thus, the electrons emitted by a VVV process can have a maximum kinetic energy of $\sim 15\,\text{eV}$. However, for Cu, the width of the valence band with appreciable density of states is only $\sim 4\,\text{eV}$ (ref. 13) and hence, a VVV Auger transition does not result in measurable emission of electrons into the vacuum.

In addition to the low-energy peak corresponding to the VVV Auger transition, the TOF spectrum of SLG on Cu (Fig. 1) shows an Auger peak corresponding to KVV transitions to the Carbon 1s level at 263 eV. The spectrum also shows the presence of adsorbed oxygen on the surface of SLG through the Auger peak corresponding to the KVV transition (503 eV) in oxygen. The spectrum from the clean polycrystalline Cu substrate after the removal of the graphene layer has three main features: the Auger peaks corresponding to $M_{2,3}VV$ and $M_1VV$ transitions at 60 and 108 eV, respectively, and a low-energy tail (LET) which has contributions from inelastic loss of the higher energy Auger peaks. The intensity under the LET (defined till 30 eV) of clean Cu is only $\sim 1.7$ times the intensity in the Cu spectrum above 30 eV. A similar LET, with intensity proportional to the intensity in the high energy Auger peaks, is expected in the electron spectrum from SLG on Cu due to the inelastic loss of higher energy Auger peaks (for example, the KVV peak from C). However, the integrated intensity in the energy range from 0 to 11 eV is more than an order of magnitude larger than the integrated intensity of the higher energy Auger peaks. Consequently, contribution from the inelastic scattering of the high energy Auger electrons is minimal in the region of the VVV peak.

**Theoretical calculation of the VVV Auger spectrum.** Quantitative information regarding the VVV Auger process was obtained by comparing the experimentally obtained VVV Auger electron energy spectrum with the theoretical spectrum. The calculation of the Auger line shape is based on a model similar to the one used for Auger neutralization of ions on solid surfaces[14]. First, the electronic structure and the positron state for free standing SLG and for SLG on Cu (111) (unit cell used to simulate SLG on a Cu (111) surface is shown in Supplementary Fig. 1) were calculated from first principles (details in the

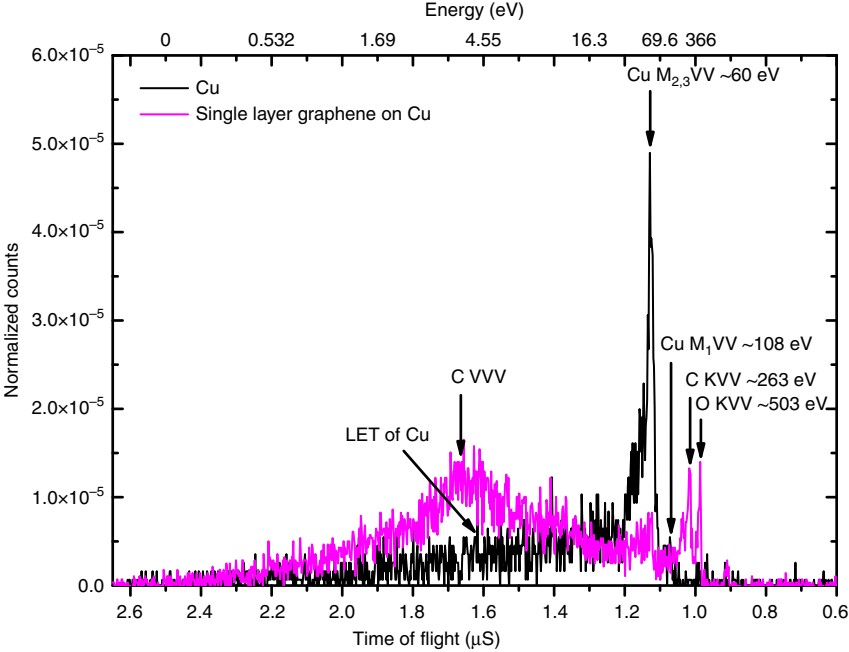

**Figure 1 | Time of flight positron annihilation-induced Auger electron spectrum.** TOF spectrum of electrons emitted from a single layer of graphene (magenta) on a polycrystalline Cu substrate following the creation of holes via positron annihilation. Also shown is the spectrum of electrons emitted from the polycrystalline Cu substrate (black) after removal of the graphene layer. The top axis shows the energy of the electrons calculated from their TOFs. Analysis in this paper shows that the strong peak at $\sim 4$ eV ($\sim 1.65$ µs) in the spectrum from graphene corresponds to emission of electrons as a result of an Auger transition in which the energy is provided by the filling of a deep hole in the 20 eV wide valence band of the graphene layer created by matter–antimatter annihilation (Fig. 2). This peak is notably absent in the spectrum taken from the clean Cu substrate, consistent with the fact that the Auger electrons resulting from the filling of holes in the relatively shallow valence band of Cu do not have enough energy to leave the surface. The TOF-PAES spectrum of the graphene also shows peaks corresponding to the Auger relaxation of core holes in C (KVV), adsorbed O (KVV) and in Cu (MVV) and the spectrum from clean Cu substrate show peaks corresponding to the $M_{2,3}$VV and $M_1$VV Auger peaks. The use of an incident 1.25 eV positron beam to create the holes eliminates the normally overwhelming beam induced secondary electron background that prevents other photon or electron based techniques from making a direct measurement of the valence Auger process.

Methods section and in Supplementary Note 1). For both free standing SLG and SLG on Cu (111), we find that the positron is located in a surface localized state (with the beam intensity used in our experiment, the probability of a second positron arriving before the annihilation of the first one is very small, <1 p.p.m., and hence there is only one positron on the surface at any time). In the case of SLG on Cu, the positron is predominantly located at the vacuum side of the graphene layer, but there is still a non-negligible overlap of the surface localized positron wave function with the first atomic layer of the Cu substrate (Fig. 3a,b). This is consistent with the experimental data for SLG on Cu which shows a small peak corresponding to the $M_{2,3}$VV Auger transition in Cu.

After calculating the electronic structure and the ground-state positron wave-function on free standing graphene, the partial annihilation rates of the positron with the valence electrons were obtained using the overlap integral of the electronic and positronic ground state wave functions. The spectrum of annihilation-induced holes, as shown in Fig. 3c, was then calculated by summing the partial annihilation rates of the positron with electronic states at a given energy. The energy distribution of the annihilation-induced holes was then used together with the valence-band density of states and the density of unoccupied states of the free standing graphene to calculate the energy spectra of electrons emitted through the VVV Auger process (equation (2) in Methods). In the calculation, it was assumed that all the valence-band holes relaxed through energetically allowed VVV transitions (the calculated energy spectrum, the calculated TOF spectrum and the electron energy

spectrum after considering the instrumental broadening are shown in Supplementary Figs 2–4, respectively). The small contribution from the LET of the high-energy Auger peaks has been subtracted from the spectral intensity below 11 eV in the experimental curve (Supplementary Note 2 and Supplementary Figs 5–7) before comparing it to the calculated spectrum. The comparison of the experimental and the theoretical VVV Auger line shapes (Fig. 4) shows that the calculated spectrum reproduces the overall width and shape of the experimental peak.

**Branching ratio of the VVV Auger process.** After correcting for the loss in the Auger peak intensities due to inelastic scattering (Supplementary Note 3), it was found that the measured ratio of the integrated intensity of the C VVV Auger peak to the integrated intensity of the C KVV Auger peak is $21 \pm 4$ (see Supplementary Note 4 for details of calculation). To estimate the branching ratio for the 'VVV' Auger decay of the deep valence hole, we have compared this measured ratio to the theoretically calculated ratio. The integrated intensity of the C KVV Auger peak was theoretically calculated by assuming that all core holes created via positron–electron annihilation decay through an Auger process which is consistent with the previous results on the Auger transition rates of K-shell holes in C (ref. 15). The ratio of the calculated integrated intensities of the C VVV and C KVV Auger peaks is 20.7, assuming all of the valence-band holes that are deep enough to result in electron emission into the vacuum undergo a VVV Auger transition. The calculated ratio is comparable to the experimental ratio if the branching ratio in

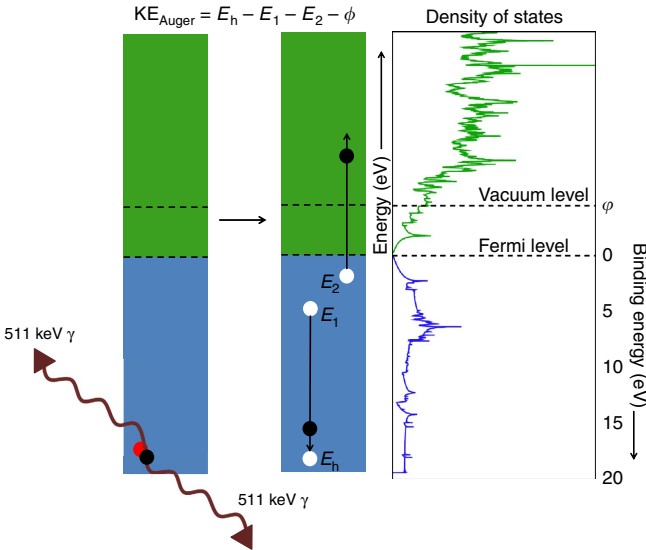

**Figure 2 | Schematic representation of the VVV Auger transition.** In the first step, a deep hole in the valence band is created by the annihilation of a valence electron with a surface bound positron (first box on the left). In the second step, an electron from a higher level in the valence band falls into the deeper hole and the energy associated with this transition is coupled to another valence electron. Electron emission into the vacuum can occur if the transition results in sufficient energy to overcome the work function. The kinetic energy of the emitted Auger electron is $KE_{Auger} = E_h - E_1 - E_2 - \phi$ where $E_h$, $E_1$ and $E_2$ are the binding energies of the electrons involved in the transition (referenced to the Fermi level) and $\phi$ is the electron work function of the single layer graphene (the middle box). A calculation of the DOS of free standing single layer graphene (far right) shows that the valence band is $\sim 20\,eV$ deep and that a significant fraction of the allowed VVV Auger transitions can result in electrons with the required energy to escape into the vacuum.

the calculation was taken to be between 0.8 and 1 (where the range of the estimate was set by the uncertainty in the measurements of the integrated intensity ratio). Therefore, nearly all deep holes (between 80 and 100%) in the valence band of graphene decay through an Auger process resulting in hole multiplication near the Fermi level. While other mechanisms, for example phonons, may contribute to line shape broadening, our data indicate that the majority of the energy associated with the initial deep hole contributes to an Auger process. Note that the higher intensity of the positron induced VVV Auger peak in comparison to the KVV Auger peak is primarily due to the fact that $\sim 99\%$ of the positrons annihilate with graphene valence electrons. Also, note that the momentum of the electron emitted via an Auger process is taken into account in the theoretical calculation of the line shapes and the integrated intensities of the Auger peaks via the phenomenological approach introduced by Hagstrum et al.[14].

## Discussion

Our work reveals that the emission of electrons as a result of Auger transitions involving the filling of deep holes in the valence band is highly efficient in graphene, with a branching ratio close to unity. Hence, VVV Auger processes can be expected to make an important contribution to the low-energy photoelectron or secondary electron spectra from graphene. The analysis of the intensity and shape of the low-energy photoelectron and secondary electron spectra from other wide band materials should consider the VVV Auger process provided the branching

ratio of the VVV Auger process is significant. The use of low-energy positrons to create valence holes in solids can be easily extended to investigate the ionization processes following the creation of deep valence holes in molecules and molecular clusters (water, Ne clusters and organic molecules)[7,8,16] by depositing them on a substrate. The experiments on organic molecules can elucidate the role of valence Auger processes in radiation damage by ionizing radiation. The efficient decay of valence hole with the emission of a low-energy electron ($<11\,eV$) in C, as shown in this paper, can be an important pathway in radiation damage of living tissues. Low energy electrons ($<20\,eV$) have been shown to be very effective in causing DNA strand breaks through dissociative electron attachment[17–19].

The modelling of the line shape of the VVV Auger peak in SLG is in good agreement with the experimental data without considering the hole–hole repulsion in the two-hole final state. However, in systems where the hole–hole repulsive potential is comparable to the width of the valence band[20,21], the correlation effects would have an important role in determining the line shape and the peak position of the VVV Auger peak. The VVV line shape analysis can thus yield key information on hole-hole correlation effects and on the density of states of the top layer of the samples. The top layer sensitivity is due to the fact that positrons deposited at low energies become trapped in a surface localized state on a two-dimensional (2D) material, where they annihilate predominantly with atoms in the topmost atomic layer (for example, in our calculations shown in Fig. 3a,b, 92% of the positrons annihilate with the single layer of graphene). The use of a low-energy positron beam thus provides a novel method to selectively probe 2D materials on a substrate and paves the way for the direct comparison of theoretical models of the density of states and correlation effects in 2D materials.

We have shown that a model used for the Auger neutralization involving the filling of a hole in an incident ion can also predict the decay of the hole in the valence band of a homogeneous material, and the model has allowed us to estimate the branching ratio for the Auger decay of valence holes in graphene. The finding that the Auger decay of the valence hole happens more than 80% of the time correlates well with earlier results on the prominence of the Auger-like processes in the graphene valence band resulting in hole multiplication[22]. Extending our measurements to other materials (like Si) will allow us to test theories of Auger relaxation in the valence band that are important in modelling fluorescence droop[23] and charge multiplication[24] in semiconductors. Our results open up new avenues for the measurement of both the intensity and electron energy distribution from an important mechanism for low-energy electron emission in solids and molecules that, up till now, has been inaccessible to direct measurement.

## Methods

**TOF-PAES.** The measurements were performed using the UT Arlington TOF-PAES system, consisting of a low energy positron beam equipped with a $\sim 1$ m flight path TOF spectrometer (Supplementary Fig. 8). The system was configured so as to permit both the transport of a very low-energy positron beam (Supplementary Fig. 9) to the sample and the measurement of the energy of electrons emitted from the sample over a wide range of energies (0.25–800 eV). The apparatus and low-energy settings used have been described in detail in Supplementary Note 5. The TOF spectra were converted to energy spectra using an energy conversion function empirically derived from a set of calibration curves (Supplementary Figs 10 and 11) as described in Supplementary Note 6 and Supplementary Fig. 12. All of the spectra shown in this paper have been normalized by dividing by a number proportional to the number of positrons annihilating at the sample as determined from the detected annihilation gamma rays.

**Sample.** SLG on Cu substrate grown using the chemical vapor deposition (CVD) technique was purchased from ACS materials. The sample was put into the UHV TOF-PAES spectrometer in the as-received condition. To obtain a PAES spectrum

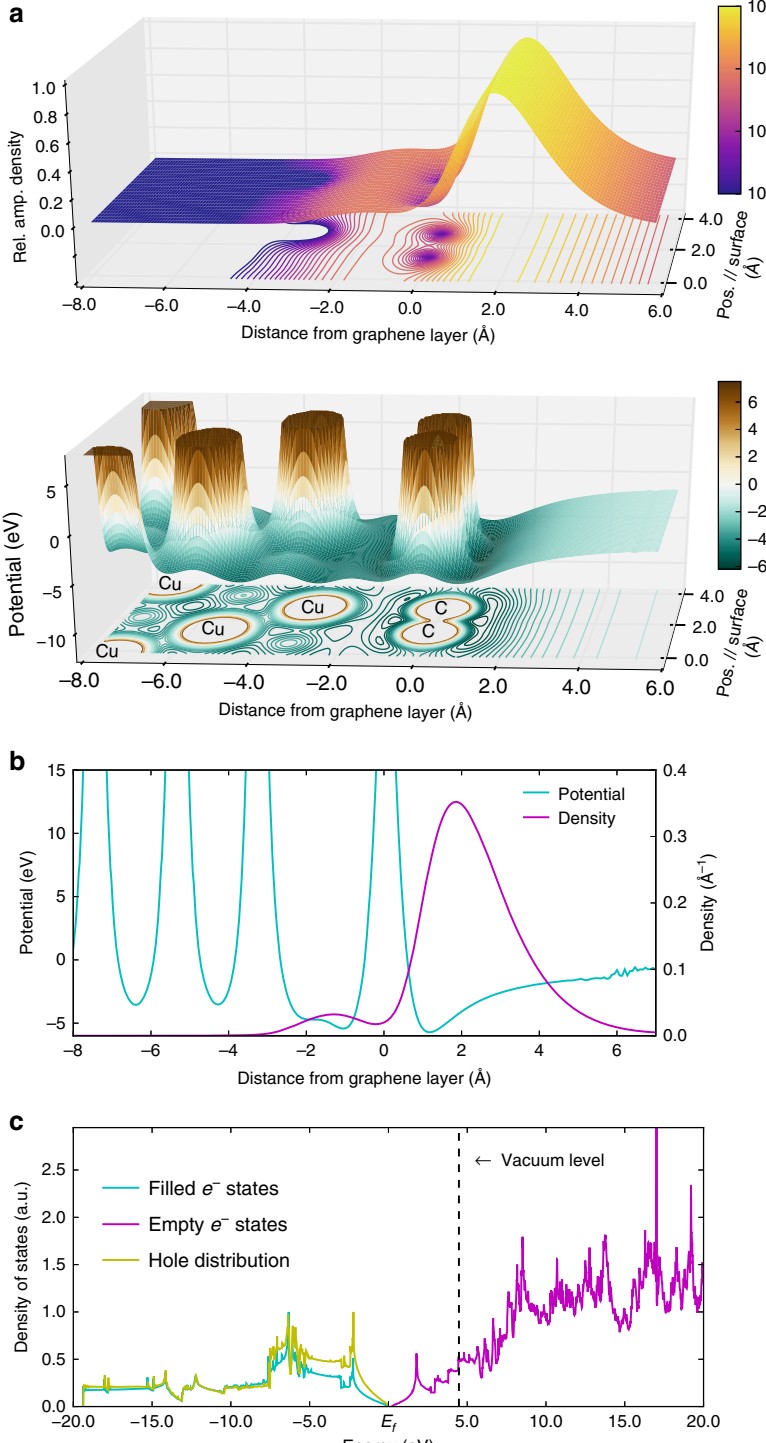

**Figure 3 | Results of theoretical calculations. (a)** Cut through the ground-state positron probability density (top panel) and the potential experienced by the positron near the top atomic layers (bottom panel) along a C–C bond on single layer graphene (SLG) on a Cu (111) substrate (The (110) plane of the unit cell is shown in the Supplementary Fig. 1). Our result shows that the positron is localized predominantly in its image potential well at the vacuum side of the graphene layer, giving signals almost exclusively from the SLG. The 2D contour plot of the potential in the bottom panel shows the position of the Cu and C atoms. **(b)** One-dimensional plot of the potential (turquoise) and the positron density (purple) averaged over the xy plane. It can be seen in this plot that the positron overlap with Cu atoms is small compared with the overlap with the C atoms. As a consequence, most of the annihilation and annihilation induced signal is due to the graphene layer consistent with our experimental results. **(c)** Density of the electronic valence (turquoise) and conduction (purple) states. The vacuum level is indicated by the dashed line. The yellow curve shows the calculated distribution of annihilation induced holes that initiate the VVV Auger process (the distribution of electronic valence states and the distribution of hole states have been scaled to agree at the peak). The calculations show that the distribution of annihilation-induced holes closely resembles the valence band density of states due to the fact that the positron partial annihilation rate is relatively constant for states in the valence band.

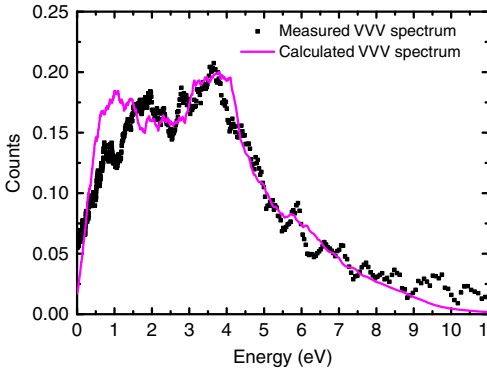

**Figure 4 | Experimental and calculated VVV Auger spectrum.**
Comparison of the measured (square) and calculated energy distribution
(line) of electrons emitted via annihilation-induced VVV Auger transition
from a single layer of graphene on Cu. The calculated VVV Auger electron
energy spectrum for a free standing graphene layer has been broadened
using a simulated instrumental response function of the TOF-PAES
spectrometer system.

from pure Cu, the SLG over-layer was sputtered using Ar ions at a pressure of
$5 \times 10^{-6}$ Torr with a sputtering current of $<1\,\mu A$. A sputtering of 4 min was
enough to obtain the $M_{2,3}VV$ and $M_1VV$ Auger peaks from Cu which show that
the SLG acts like an excellent passivation layer. The clean Cu data shown in this
paper has been sputtered for 20 min.

**Data analysis.** The LET arising out of inelastic loss processes and cascade
processes of higher energy Auger peaks in the TOF-PAES spectra of SLG was
subtracted from the low-energy VVV Auger peak. The LET of the Cu PAES
spectrum was fit to a model function (Supplementary Fig. 5) and was used to
estimate the LET in the TOF-PAES spectrum of SLG by scaling the LET intensity
to the spectral intensity from 30 eV to 750 eV (which encompasses the Auger
peaks of C, O and Cu) in SLG. The subtraction process is detailed in
Supplementary Note 2 and Supplementary Fig. 6.

The TOF spectrometer used in the experiment was simulated using SIMION 8.1
(ref. 25) to explore the energy dependence of the electron transport efficiency of the
spectrometer. A drop in transport efficiency was found for electron energies
above 500 eV and for energies below 1 eV (Supplementary Fig. 13). Transport
efficiency as a function of energy was fit to a functional form and was used to
correct the Auger spectra (Supplementary Note 7). To account for instrumental
broadening, the calculated Auger spectrum has been used as an input to the
charged particle trajectory simulation of the experimental setup (Supplementary
Note 1 and Supplementary Fig. 2). The output of the simulation is a TOF
spectrum (Supplementary Fig. 3) which has been converted to the energy
spectrum (Supplementary Fig. 4) and smoothed using the same algorithm
(Supplementary Note 6) employed on the experimental TOF spectrum.

**Theoretical calculations.** The line shape of the annihilation induced VVV
transition is obtained by first calculating the distribution of annihilation induced
holes in the valence band and then applying formalism similar to that used
previously in the interpretation of ion neutralization experiments[14,26].

The distribution of annihilation induced holes in the valence band was
determined by first solving for the positron surface state wave function, and then
using this wave function to calculate the partial annihilation rate $\lambda(\epsilon_h)$ of the
positron with electron states at energy $\epsilon_h$. The calculation of the surface state wave
functions and the annihilation rates, $\lambda$, with core and valence electrons were
performed within the framework of the two-component electron–positron density
functional theory. We use a non-local weighted density approximation to describe
the electron-positron correlation effects[27], using the Drummond enhancement
factor[28], to obtain a correct image potential far away from the surface. The
distribution of initial, annihilation-induced holes are determined by the partial
annihilation rate $\lambda(\epsilon_h)$ of the positron with the electronic states at a given energy
$\epsilon_h$. This quantity is obtained from an integration of the annihilation rate over the
Brillouin zone with each electronic state.

The model VVV Auger energy distribution, $A(\epsilon_{esc})$, is then calculated as:

$$A(\epsilon_{esc}) = \int d\epsilon_h \frac{\lambda(\epsilon_h)D_c(\epsilon_{esc} + \epsilon_{vac})P_e(\epsilon_{esc})T(\epsilon_h, \epsilon_{esc} + \epsilon_{vac})}{N(\epsilon_h)} \quad (2)$$

where $\epsilon_{esc}$, is the kinetic energy of the Auger electron after it has exited the surface,
$D_c$ is the density of unoccupied states, $P_e(\epsilon_{esc})$ is the escape probability function[26]

and $T(\epsilon_h, \epsilon_{esc} + \epsilon_{vac})$ is the Auger transform[29]. The normalization $N(\epsilon_h)$ expresses
that each hole decays through an Auger transition.

The Auger transform uses the typical self-convolution of the one-particle
density of states to describe the transition of the initial, annihilation induced,
one-hole state to the final two-hole state[29,30]. This type of convolution integral
has been successful in describing band-like CVV Auger spectra[30]. More details of
the calculation are given in Supplementary Note 1.

**Data availability.** The data that support the findings of this study are available
from the corresponding author upon reasonable request. The TOF-PAES data, the
experimental VVV energy spectrum and the calculated VVV energy spectrum can
be obtained from https://uta-ir.tdl.org/uta-ir/handle/10106/26650.

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

## Acknowledgements

The experiments in this work were supported by the grant NSF DMR 1508719. A.H.W and A.R.K. gratefully acknowledge support for the building of advanced positron beam through the grant NSF DMR MRI 1338130. V.C. and R.S. were supported by the FWO-Vlaanderen through Project No. G. 0224.14N. The computational resources and services used in this work were in part provided by the VSC (Flemish Supercomputer Center) and the HPC infrastructure of the University of Antwerp (CalcUA), both funded by the Hercules Foundation and the Flemish Government (EWI Department). The work at Northeastern University was supported by the US Department of Energy (DOE), Office of Science, Basic Energy Sciences grant number DE-FG02-07ER46352 (core research), and benefited from Northeastern University's Advanced Scientific Computation Center (ASCC), the NERSC supercomputing center through DOE grant number DE-AC02-05CH11231, and support (applications to layered materials) from the DOE EFRC: Center for the Computational Design of Functional Layered Materials (CCDM) under DE-SC0012575.

## Author contributions

V.A.C., A.R.K., K.R. and A.H.W. carried out the initial measurements and conceived of the interpretation of the results. V.C., R.S., B.B. and B.P. performed the theoretical calculations. A.J.F. modelled the instrument response, M.D.C., R.W.G., K.S., S.K.I. and A.D.M. contributed to the preparation and characterization of the samples and to the analysis of the data. All authors discussed the results and contributed to writing of the manuscript.

## Additional information

**Competing interests:** The authors declare no competing financial interests.

