## [Peer Review File · Nature Communications]

File name: Supplementary Information

Description: Supplementary Figures, Supplementary Notes and Supplementary References

File name: Peer Review File

Description:

Supplementary Note 1: Theoretical calculation of the VVV Auger spectrum

1.1 Introduction

In order to support our claims for the observation of the VVV Auger process in single layer graphene (SLG), we carried out first-principles calculations in the framework of the two-component electron-positron density functional theory (2CDFT) [1, 2]. We work in the ‘zero positron density limit’ of the formalism as we only consider perfect surfaces, where the positron will form a delocalized state. The electronic structure of the system under consideration remains, in such case, unperturbed by the presence of the positron. This allows us to first obtain the electronic ground state properties of the system and, using the ground state electron density, construct the Kohn-Sham potential for the positron.

1.2 Electronic structure

To obtain the electronic properties, we use the PAW method [3], as implemented in VASP [4, 5, 6]. The kinetic energy cutoff for the plane-wave expansion of the wave functions is set at 520 eV in all calculations. For the relaxation and ground state calculation of free standing graphene, a Γ -centered k-mesh of 15 x 15 x 1 is used, and is refined to 45 x 45 x 1 to calculate the DOS accurately. In the case of the ground state calculation and relaxation of graphene ontop of the Cu (111) surface, a Γ -centered k-mesh of 21 x 21 x 1 is selected. In all calculations, the structures are relaxed until the forces on the atoms are smaller than 5 meV/Å. A vacuum of 15 Å is present in all slab models to avoid interaction between periodic images. Electron-electron exchange and correlation effects are described using the optB88-vdW functional [7] which includes long-range Van der Waals effects and yields accurate lattice constants for both Cu [8] and Graphite [9]. We find that this functional also reproduces a graphene - Cu interface spacing of 3.287 Å, which is in agreement with the value of 3.34 ± 0.06 Å obtained from total-reflection high-energy positron diffraction experiments [10]. Supplementary figure 1 shows the unit cell used in the Cu (111) + graphene calculations.

Supplementary figure 1: **Unit cell of graphene on Cu(111)** (a) Side view and (b) top view of the unit cell used in this work to simulate graphene on a Cu (111) surface. Cu atoms are indicated in blue and C atoms in brown and gray, where, for clarity, C atoms at the different Cu surfaces have different colors.

1.3 Positron state

Positron calculations are performed with the MIKA/Doppler package [11] in which a superposition of free atomic core densities, i.e. charge and potential, are added to the self-consistent valence properties obtained from the electronic calculations. In order to accurately describe the positron surfaces state; we use the weighted density approximation (WDA) as described in Ref. [12]. The WDA takes into account the non-local screening effects which are important to obtain the correct asymptotic '1/z' behaviour for the positron potential in the vacuum region. In the WDA for electron-positron calculations, one can perform a shell partitioning to take

into account that electrons that occupy different shells of an atom participate differently in the screening of the positron charge [13]. The partitioning employed in this work is different from the one described in Ref. [13] and is instead inspired by the core-valence partitioning described in Refs [14, 15]. More specifically, we describe the screening by the core electron in a local density approximation and the screening by valence electrons in the WDA approach. The formula for the electron-positron correlation potential is then given by

$$V_c(\mathbf{r}_p) = \frac{n_c^e(\mathbf{r}_p)}{n^e(\mathbf{r}_p)} \epsilon_c^{e-p}[n^e(\mathbf{r}_p)] - \int_0^1 dQ \int d\mathbf{r}_e \frac{\Delta n_c^{e-p}(\mathbf{r}_e|\mathbf{r}_p; n^*(\mathbf{r}_p); Q)}{|\mathbf{r}_p - \mathbf{r}_e|} \quad (1)$$

with $\epsilon_c^{e-p}[n^e(\mathbf{r}_p)]$ being the LDA correlation potential. The induced charge density due to the presence of the positron at \mathbf{r}_p , i.e. the screening cloud, is determined by:

$$\Delta n_c^{e-p}(\mathbf{r}_e|\mathbf{r}_p; n^*(\mathbf{r}_p); Q) = n_v^e(\mathbf{r}_e) \left[g_{n^*(\mathbf{r}_p)}^h(|\mathbf{r}_p - \mathbf{r}_e|; Q) - 1 \right] \quad (2)$$

where we take following form for the electron-positron pair-correlation function [14]

$$g_{n^*(\mathbf{r}_p)}^h(|\mathbf{r}_p - \mathbf{r}_e|; Q) = 1 + \{\gamma[n^*(\mathbf{r}_p)] - 1\} e^{-a[n^*(\mathbf{r}_p)]|\mathbf{r}_p - \mathbf{r}_e|} \quad (3)$$

with

$$a^3[n^*(\mathbf{r}_p)] = 8 \pi n^*(\mathbf{r}_p) \{\gamma[n^*(\mathbf{r}_p)] - 1\} \quad (4)$$

For the enhancement factor γ , we take recent parameterization of Ref. [16]. Note that this correlation function assumes that the induced electron charge scales linearly with the charge of the positron Q in the adiabatic integral for V_c . As a consequence, this integral can be performed analytically and results in a factor 1/2 in front of the non-local part of the potential. The effective electron density at the positron position $n^*(\mathbf{r}_p)$ is obtained by imposing the modified sum rule

$$\int d\mathbf{r}_e \Delta n_c^{e-p}(\mathbf{r}_e|\mathbf{r}_p; Q; n^*(\mathbf{r}_p)) = Q \frac{n_v^e(\mathbf{r}_p)}{n^e(\mathbf{r}_e)} \quad (5)$$

In the above formulae, n^e , n_v^e and n_c^e denote the total, valence and core electron densities, respectively. The electrons we consider as valence in our calculations are the C(2s), C(2p), C(3d) and C(4s) shells.

1.4 Auger spectra

To calculate the Auger spectra, we base our model on the work by Hagstrum on Auger neutralization [17, 18]. Once a hole is created in the valence band at an energy ϵ_h , an electron from a higher lying electronic level ϵ_1 can drop in this hole by transferring the energy difference between the two levels $\epsilon_h - \epsilon_1$ to a second electron at an energy level ϵ_2 . The transition rate of this process is assumed to be proportional to the density of states (DOS) at the initial energies, $D_v(\epsilon_1)$ and $D_v(\epsilon_2)$. The relative amplitude of the escaping electron at a given energy ϵ_{esc} is then obtained from a self-convolution of the DOS with inclusion of the escape function, i.e. the Auger transform $T[\epsilon_h, \epsilon_{vac} + \epsilon_{esc}]$, multiplied with the amount of available final states $D_c(\epsilon_{vac} + \epsilon_{esc})$ and with the escape function $P_e(\epsilon_{esc})$. The self-convolution of the DOS takes into account all possible combinations of initial energies ϵ_1 and ϵ_2 which can lead to the emission of a Auger electron with a kinetic energy of ϵ_{esc} [19], whereas the escape function is a geometric factor that takes into account the fact that Auger electron needs sufficient kinetic energy perpendicular to the surface to escape to the vacuum. The distribution of initial, annihilation induced, holes is determined by the partial annihilation rate $\lambda(\epsilon_h)$ of the positron with the electronic states at a given energy ϵ_h . The resulting expression for the Auger spectrum is given by

$$A(\epsilon_{esc}) = \int d\epsilon_h \frac{\lambda(\epsilon_h) D_c(\epsilon_{vac} + \epsilon_{esc}) P_e(\epsilon_{esc}) T[\epsilon_h, \epsilon_{vac} + \epsilon_{esc}]}{N(\epsilon_h)} \quad (6)$$

where the Auger transform is given by

$$T[\epsilon_h, \epsilon_{vac} + \epsilon_{esc}] = \int_{-\infty}^{\epsilon_v} d\epsilon_1 \int_{-\infty}^{\epsilon_v} d\epsilon_2 D_v(\epsilon_1) D_v(\epsilon_2) \delta((\epsilon_1 - \epsilon_h) + (\epsilon_2 - \epsilon_{vac} - \epsilon_{esc})) \Theta(\epsilon_1 > \epsilon_h) \quad (7)$$

Here, δ and Θ denote the Dirac δ -function and the Heaviside function, respectively. The top of the valence band is denoted by ϵ_v and the vacuum level by ϵ_{vac} . The kinetic energy of the escaping electrons ϵ_{esc} is thus measured w.r.t. the vacuum level. The escape function that models the influence of the momentum of the electron on the escape probability is give as

$$P_e(\epsilon_{esc}) = \frac{1}{2} \left[1 - \left(\frac{\epsilon_{vac} - \epsilon_{(V)}}{\epsilon_{vac} + \epsilon_{esc} - \epsilon_{(V)}} \right)^\beta \right]^\alpha \quad (8)$$

where $\epsilon_{(V)}$ is the average value of the electron potential in the bulk of the sample. The parameter β were chosen to be 0.5. The parameter α was varied from 0 to 1. It was

found that $\alpha = 0.25$, produced the best match between the theoretical spectrum and the experimentally measured VVV Auger peak. The value of the parameter found is close to the values reported in [17]. In our model, we assume that all initial (annihilation induced) hole are relaxed through an Auger transition and neglect any higher order contributions to the Auger spectrum. Note that not every Auger transition automatically results in the emission of an electron from the sample. A fraction of the electrons end up in the conduction band below the vacuum energy and a certain fraction cannot overcome the surface barrier because of the angle of emission. The normalization factor in the Auger spectrum expression is given by

$$N(\epsilon_h) = \int_{\epsilon_c}^{\infty} d\epsilon_f D_c(\epsilon_{\text{vac}} + \epsilon_f) T[\epsilon_h, \epsilon_{\text{vac}} + \epsilon_f] \quad (9)$$

where ϵ_c denotes the bottom of the conduction band and ϵ_f the energy of the final state of the Auger electron. The annihilation rate of the positron with the electronic states at ϵ_h is obtained by an integral over the Brillouin zone

$$\lambda(\epsilon_h) = \frac{1}{\Omega} \sum_i \int_{\Omega} d\mathbf{k} \lambda_i(\mathbf{k}) f(\epsilon_i(\mathbf{k})) \delta(\epsilon_h - \epsilon_i(\mathbf{k})) \quad (10)$$

where the sum runs over all electronic states, i denotes a set of band and spin indices and $f(\epsilon_i(\mathbf{k}))$ gives the occupation of the electronic states. The volume of the Brillouin zone is given by Ω . The partial annihilation rates are obtained from

$$\lambda_i(\mathbf{k}) = \pi r_e^2 c \int d\mathbf{r} |\psi_{i,\mathbf{k}}^-(\mathbf{r})|^2 |\psi^+(\mathbf{r})|^2 \gamma(n_c^e(\mathbf{r}) + n^*(\mathbf{r})) \quad (11)$$

Above, $\pi r_e^2 c$ is the annihilation rate constant [20], with r_e the classical electron radius and c the speed of light. The electronic one-particle and positronic ground-state wave functions are denoted by $\psi_{i,\mathbf{k}}^-(\mathbf{r})$ and $\psi^+(\mathbf{r})$, respectively.

1.5 Instrumental broadening

The calculated VVV Auger electron energy spectrum was broadened to take into account the broadening of the spectrum due to the instrumental resolution function. To calculate the effect of the instrument resolution, trajectories of electrons, whose energies were distributed according to the calculated VVV spectrum (Supplementary figure 2), were obtained using the SIMION 8.1 ® [21] simulation of the instrument. The output of the simulation was a histogram of the TOF's of the electrons (Supplementary figure 3) which is equivalent to the TOF spectrum

obtained in the experiment. The simulated TOF spectrum was converted to the energy spectrum using the energy conversion method adopted for the experimental TOF spectrum. The energy spectrum obtained as an output of the SIMION 8.1 ® simulation (Supplementary figure 4) is smoothed using a forward averaging algorithm before comparing to the experimentally obtained spectrum.

Supplementary figure 2: **Calculated VVV Auger spectrum.** The VVV Auger electron energy spectrum calculated using equation (6). The inputs required to model the spectrum is obtained from first principle electronic structure and positron state calculations.

Supplementary figure 3: **Simulated TOF-Spectrum.** The TOF spectrum of electrons obtained from SIMION 8.1 ® simulations [21] of the TOF-PAES spectrometer. The input energy distribution of the electrons flown through the simulated instrument is the VVV Auger electron energy spectrum given in supplementary figure 2.

Supplementary figure 4: **Calculated VVV Auger spectrum after considering instrumental broadening.** The energy spectrum obtained by converting the TOF spectrum given in supplementary figure 3. The energy spectrum resembles the electron energy distribution that was given as input to the trajectory simulations. The instrument response function smoothes out the sharp peaks in the input distribution function as given in supplementary figure 2.

Supplementary Note 2: Low energy tail (LET) of Auger peaks

The PAES spectrum from clean Cu shows a low energy tail extending all the way to zero which is due to the inelastic loss of the 60 eV Auger electrons. Since, the occurrence of the LET is dependent on the creation of a core hole, the intensity under the LET should be proportional to the intensity under the higher energy Auger peak due to core hole relaxation. A similar LET will contribute to the intensity in the low energy region of the Auger electron spectrum from SLG. In order to subtract this contribution, the LET obtained in the Auger spectrum of clean Cu was fit to an ‘exponential modified Gaussian’ as shown in supplementary figure 5. The function was used to calculate the LET from the higher energy Auger peaks in SLG (eg. C KVV) and was subtracted from the VVV Auger peak in SLG. The intensity of the LET that was subtracted is proportional to the intensity in the Auger spectrum of SLG from 30 eV to 750 eV. This energy range encompasses the $M_{2,3}VV$ peak from Cu, KVV peak from C and KVV peak from O. The VVV spectrum was thus modified according to the equation

$$N_{\text{Auger}}^{\text{VVV-LET}}(E) = N_{\text{Auger}}^{\text{VVV}}(E) - \frac{I_{30-750}^{\text{SLG}}}{I_{30-750}^{\text{Cu}}} \text{LET}(E) \quad (12)$$

Here I_{30-650}^{SLG} is the intensity in the PAES spectrum of SLG from 30 eV to 750 eV and

I_{30-750}^{Cu} is the intensity from 30 eV to 750 eV in the PAES spectrum of clean Cu. $LET(E)$ is the exponential modified Gaussian function obtained from the fit to the low energy Auger spectrum of clean Cu, $N_{\text{Auger}}^{\text{VVV}}(E)$ is the measured VVV Auger electron energy spectrum and $N_{\text{Auger}}^{\text{VVV}-LET}(E)$ is the spectrum after subtracting the LET. The VVV Auger peak before and after LET subtraction is shown in supplementary figure 6.

In a similar way, while calculating the intensity of the C KVV peak, the contribution to the C KVV peak intensity from the inelastic loss of the Auger electrons that has energy higher than the C KVV peak has to be subtracted. In supplementary figure 7 the C KVV and O KVV peaks from SLG are shown along with the O KVV peak obtained from the surface of clean Cu. To obtain a Cu spectrum with an O KVV peak, a freshly sputtered clean polycrystalline Cu sample was exposed to O at a pressure of 5×10^{-7} Torr for 6600 seconds. Please note that the intensity under the O KVV peak obtained from Cu sample has been normalised to the intensity under the O KVV peak obtained from SLG. The inelastic tail of the O KVV peak extending to the C KVV region was thus estimated by using the oxygen peak in the clean Cu PAES spectrum. The O KVV loss tail has been subtracted to get an accurate intensity of the C KVV peak in the Auger spectrum of SLG.

Supplementary figure 5: **Fit to the low energy tail of Cu.** The PAES spectrum obtained from clean Cu shows the $M_{2,3}VV$ 60 eV Auger peak and the low energy tail of the peak (LET) that has contributions from the inelastic loss of the 60 eV electrons in the sample. The intensity under the LET is proportional to the intensity under the $M_{2,3}VV$ Auger peak as its occurrence depends on the creation of a core hole and the emission of a $M_{2,3}VV$ Auger electron. The LET from 0 to 30 eV is fit to an exponential sided Gaussian (magenta) and is used to subtract a similar LET from the low energy electron spectra from SLG.

Supplementary Figure 6: **Subtraction of low energy tail from VVV.** The VVV Auger peaks before (blue) and after (black) subtracting the LET (magenta) from higher energy Auger peaks.

Supplementary figure 7: **Subtraction of the low energy tail from C KVV Auger peak.** The C KVV Auger peak before (square) and after (triangle) subtracting the inelastic loss tail (circle) from O KVV peak.

Supplementary Note 3: Correction for the loss in the Auger peak intensities due to inelastic scattering

The transmission factor of Auger electrons (T) can be calculated following the formulation of Seah et al. [22]. In electron induced Auger electron spectroscopy, the Auger current from a substrate into a solid angle $d\Omega$ at an angle θ to the surface

normal with a uniform adsorbate coverage can be given as

$$I_n^S(\theta) = I_0 A_S \exp\left(-\frac{n}{\lambda_C^S \cos(\theta)}\right) \sin(\theta) d\theta d\varphi \int_0^\infty \exp\left(-\frac{z}{\lambda_S^S \cos(\theta)}\right) dz \quad (13)$$

where n is the thickness of the condensate, z is the depth normal to the surface taken to be zero at the condensate substrate interface, λ_C^S and λ_S^S are inelastic mean free paths of substrate Auger electrons through condensate and substrate layers respectively, A_S is the Auger electron current per unit solid angle per incident electron and I_0 is the incident electron beam current. The depth at which Auger electrons are created is governed by the penetration depth of the incident electrons which is much larger than the inelastic mean free path, and hence the limit of the integration is set from 0 to ∞ . In order to get the Auger current from a clean surface, the thickness of the condensate, n , is put to 0.

In PAES, the Auger electron creation depth is determined by the extent of overlap of the surface bound positron wave function with the top atomic layer. This limits the Auger electron creation depth to the top most atomic layer and application of this result to equation (13) removes the integral over depth. Thus, the Auger electron intensity from a clean surface ($n=0$) in a PAES spectrum can be written as

$$I^S = I_0 A_S \iint_0^{\frac{\pi}{2}} \exp\left(-\frac{z}{\lambda_S^S \cos(\theta)}\right) \sin(\theta) d\theta d\varphi \quad (14)$$

where I_0 is the number of positrons in the surface state, A_S is the number of Auger electrons produced per positron in the surface state and z is half thickness of the top atomic layer of the substrate. The transmission factor, T , can be then be defined as

$$T = \frac{I^S}{I_0 A_S \iint_0^{\frac{\pi}{2}} \sin(\theta) d\theta d\varphi} = \frac{\int_0^{\frac{\pi}{2}} \exp\left(-\frac{z}{\lambda_S^S \cos(\theta)}\right) \sin(\theta) d\theta}{\int_0^{\frac{\pi}{2}} \sin(\theta) d\theta} \quad (15)$$

Equation (15) is used to calculate the transmission factor for the VVV and KVV Auger electrons and was utilised in equation to calculate the integrated intensity ratios of the VVV and the KVV Auger peaks.

Supplementary Note 4: Calculation of the efficiency of the VVV Auger process from the ratios of intensities VVV and KVV Auger peak

In order to determine the efficiency of the VVV Auger process, the intensity of the VVV Auger peak is compared to the KVV Auger peak from Carbon. By taking the ratio, the detector related parameters which are not dependent on the energy of the ejected electrons get cancelled out. This is shown in the following discussion.

Suppose $N_{\text{Auger}}(E)$ be the number of Auger electrons with energy E (due to VVV, or KVV) detected by the TOF spectrometer. These Auger electrons are produced in the sample after annihilation of N_A positrons. Considering the detector and process efficiencies we can express $N_{\text{Auger}}(E)$ for VVV Auger transition as

$$N_{\text{Auger}}^{\text{VVV}}(E) = N_A f_V A_{\text{VVV}} f_{\text{VVV}}(E) T \Omega(E) \epsilon_{511, \text{BaF}_2} \epsilon_{\text{MCP}} \eta(E) \quad (16)$$

where f_V is the fraction of positrons that annihilate with valence electrons, A_{VVV} is the efficiency of the VVV transition and $f_{\text{VVV}}(E)$ is the fraction of VVV Auger transitions that result in the emission of electrons with energy E from the valence band of graphene. T is the transmission factor of the emitted Auger electron through the sample that depends on the inelastic mean free path, Ω is the Auger electron emission factor that depends on the Auger electron emission angle, $\epsilon_{511, \text{BaF}_2}$ & ϵ_{MCP} are the detector efficiencies of the BaF_2 and MCP respectively and $\eta(E)$ is the electron transport efficiency of the TOF spectrometer. f_{VVV} , T and Ω are considered to be independent of each other. $\eta(E)$ has been calculated using a charged particle trajectory simulation of our spectrometer and the correction has been applied to the Auger spectrum. The emission solid angle of Auger electrons at the surface is 2π sr and hence the emission factor Ω has a maximum value of $2\pi/4\pi = 0.5$ if all Auger electrons emitted into 2π sr overcome the surface dipole barrier. An energy dependent empirical function models the influence of the angle of emission of the Auger electron on the probability of overcoming the surface barrier. The front plate of the MCP is biased to 200 V and ϵ_{MCP} is approximately constant for electrons with energies from 200 eV to 800 eV. The Auger electron spectrum is normalised to the number of gamma counts as detected by the NaI scintillation detector. The intensity of the VVV peak is then found by integrating the $N_{\text{Auger}}^{\text{VVV}}(E)$ over the required energy range and can be written as

$$I_{\text{VVV}} = \frac{\int_{E_1}^{E_2} (N_{\text{Auger}}^{\text{VVV}}(E)/\eta(E)) dE}{N_\gamma} = \frac{N_A \epsilon_{511, \text{BaF}_2} \epsilon_{\text{MCP}} T \int_{E_1}^{E_2} f_V f_{\text{VVV}} \Omega_{\text{VVV}} A_{\text{VVV}} dE}{N_\gamma} \quad (17)$$

$N_{\text{Auger}}^{\text{VUV}}(E)/\eta(E)$ is the TOF-PAES spectrum corrected for the energy dependent transport efficiency of the TOF-PAES spectrometer. Using similar arguments the intensity of KVV Auger peak from the same sample can be expressed as

$$I_{\text{KVV}} = \frac{\int_{E_3}^{E_4} (N_{\text{Auger}}^{\text{KVV}}(E)/\eta(E)) dE}{N_{\gamma}} = \frac{N_{\text{A}} \epsilon_{511, \text{BaF2}} \epsilon_{\text{MCP}} T' \int_{E_3}^{E_4} f_{\text{K}} f_{\text{KVV}} \Omega_{\text{KVV}} A_{\text{KVV}} dE}{N_{\gamma}} \quad (18)$$

$f_{\text{KVV}} = 1$ in equation (18), as KVV Auger transition will always result in electrons with energy greater than the work function of the sample. It can be reasonably assumed that the emission factor Ω_{KVV} of C KVV Auger electrons (~ 263 eV) through SLG is independent of energy. $f_{\text{V}} = \lambda_{\text{V}}/\lambda$ and $f_{\text{K}} = \lambda_{\text{K}}/\lambda$, where λ_{V} and λ_{K} are annihilation rates of positrons with valence electrons and 1s (K) electrons respectively. λ is the total annihilation rate of positrons in the sample (SLG). By taking the ratio of I_{VUV} to I_{KVV} and rearranging transmission factors we get

$$\frac{I_{\text{VUV}} T'}{I_{\text{KVV}} T} = \frac{\int_{E_1}^{E_2} f_{\text{V}} f_{\text{VUV}} \Omega_{\text{VUV}} A_{\text{VUV}} dE}{\int_{E_3}^{E_4} f_{\text{K}} \Omega_{\text{KVV}} A_{\text{KVV}} dE} \quad (19)$$

The integral, $\int_{E_1}^{E_2} f_{\text{V}} f_{\text{VUV}} \Omega_{\text{VUV}} A_{\text{VUV}} dE$, is equal to the integrated intensity of the calculated VUV Auger electron energy spectrum and $\int_{E_3}^{E_4} f_{\text{K}} \Omega_{\text{KVV}} A_{\text{KVV}} dE$ is the integrated intensity of the calculated KVV Auger electron energy spectrum. Thus, by comparing the experimentally calculated intensity ratio to the theoretically calculated intensities, it is possible to get an estimate of the efficiencies of the VUV Auger processes if the efficiency of the KVV process is known.

Supplementary Note 5: Time of Flight – Positron Annihilation induced Auger electron Spectrometer (TOF-PAES)

The Auger electron spectra from single layer graphene on a polycrystalline Cu substrate and from clean polycrystalline Cu were collected using the TOF-PAES spectrometer at UT, Arlington [23]. The spectrometer consists of a low energy positron beam and a TOF system with a 1 m flight path (L) for measuring the energy of electrons ejected from the sample as shown in supplementary figure 8. A magnetic field of 0.0040 Tesla was used to confine incoming positrons and outgoing electrons to the beam axis. A permanent magnet placed behind the sample creates a field of 0.045 Tesla on the sample surface. This creates a magnetic field gradient that increases the velocity component of the electrons parallel to beam axis ($v_{||}$) and helps to detect all the electrons ejected into 2π sr. A schematic of the TOF spectrometer is shown in Figure 1. A beam of low energy positrons are bent around the micro channel plate (MCP) electron detector and are brought back into the beam axis using

Supplementary figure 8: **Schematic of the TOF-PAES spectrometer.** The trajectory of the incoming positrons (red arrow) and that of the outgoing electrons (black arrow) through the TOF-PAES spectrometer. The positron induced signals collected during the experiment are the Auger electrons emitted as a result of electron-positron annihilation and the two 511 keV annihilation gammas. The time difference between the signals from BaF₂ detector and the electron detector is used to produce the TOF spectrum of electron and the gamma spectrum from NaI is used to normalise the corresponding TOF spectrum. An Auger process initiated by the annihilation of a positron with a core electron results in emission of an electron with kinetic energy $KE_{Auger} = E_x - E_y - E_z - \Phi$ as shown in the schematic.

two sets of $\mathbf{E} \times \mathbf{B}$ plates as shown in the schematic. These positrons travel through a grounded TOF tube to reach the sample. The annihilation induced electrons ejected from the sample enter the TOF tube and travels most of the 1 m flight path through a field free region causing minimum variation in its velocity component parallel to the beam axis because of external electric fields. The electrons reaching the $\mathbf{E} \times \mathbf{B}$ plates are deflected into the MCP detector. The annihilation gammas are detected by a BaF_2 scintillation detector and by a NaI scintillation detector. The time difference between the detection of the annihilation gamma by the BaF_2 scintillation detector and the detection of electrons by the MCP is used to produce a histogram of the TOF's of the electron. The TOF-PAES spectrum is normalised to the total gamma counts (integrated from gamma energy of 70 keV to 588 keV) detected by the NaI scintillation detector. The kinetic energy of the incoming positrons was measured using the TOF tube as a retarding field analyser. For the measurement of beam energy the sample was biased at -250 V and the intensity of the secondary electrons measured using the TOF spectrometer was plotted as a function of retarding positive bias on the TOF tube. The derivative of this integrated spectrum gives the distribution of the kinetic energy of the positrons inside the TOF tube as shown in supplementary figure 9.

Supplementary figure 9. **Energy distribution of the positrons at the TOF tube.** To measure the energy distribution of incoming positrons, the sample is biased to -250 V with respect to the ground and the secondary electrons produced by the impinging positrons are measured at the MCP. The integrated intensity under the secondary peak is plotted as a function of positive bias on the TOF tube. It can be seen that for a bias of > 1V almost no positrons reach the sample and thus, the intensity of the secondary electron peak produced by the positron impact goes to zero. Therefore, we consider 1 eV as the maximum kinetic energy of the positrons reaching the sample. The line through the derivative counts is a Gaussian fit to the kinetic energy distribution. The peak of the fit Gaussian functions is at 0.5 eV and the FWHM is ~ 0.26 eV. Error on the integrated intensity is the standard deviation for a counting experiment and is calculated by taking the square root of the integrated counts.

Supplementary Note 6: Conversion of the TOF spectrum to energy spectrum

In the present experiment, we have measured the TOF spectrum in reverse timing method where the electron signal is used as the start signal and a delayed BaF₂ signal is used as the stop signal. The reverse TOF spectrum is collected by the multichannel analyser (MCA) in terms of channel number (ζ) that is proportional to the measured flight time. In this experiment 2048 channels of the MCA correspond to 4 μ s time window of the time to amplitude converter (TAC). Thus, the flight time, t_{flight} , corresponding to channel number, ζ can be found as

$$t_{\text{flight}} = 4 - \left(\frac{4}{2048}\right) (\zeta) \mu\text{s} \quad (20)$$

If the electrons travel from sample to electron detector in a field free region then,

$$t_{\text{flight}} = \frac{L}{v_{\parallel}} = \sqrt{\frac{mL^2}{2E}} \quad (21)$$

where m is the mass of the electron, L is the length of the flight path, E is the kinetic energy and v_{\parallel} is the velocity of the electron parallel to the beam axis. As low energy electrons spend an appreciable amount of time in the $\mathbf{E} \times \mathbf{B}$ region, there will be deviations from the relation shown by equation (21). In order to find the relation between the flight time (channel number) and the kinetic energy, the sample is biased to different voltages (0.5 V to 900 V) and the electron spectrum is collected in each case. The electron spectrum shifts towards higher channel numbers (lower flight times) with increasing sample bias as the electron travels from the sample to the analyser with additional energy corresponding to the sample bias as shown in supplementary figure 10. The channel number corresponding to the electron energy that is equal to $-e \cdot (\text{sample bias})$ is found by fitting a line to the falling edge of the secondary electron spectra as shown in supplementary figure 11. The plot between the edge channels thus found and $(\text{sample bias})^{-\frac{1}{2}}$, as shown in supplementary figure 12, is fit to a fourth order polynomial. The polynomial is used to convert the TOF (channel) spectra to the energy spectra. The count in each energy bin, $N(E)$, is found by using the counts in the channel number, ζ ($N(\zeta)$), corresponding to the energy E and the Jacobian, $\left(\frac{dE}{d\zeta}\right)$, calculated from the functional relation between the channel number and the sample bias. The relation between $N(E)$ and $N(\zeta)$ is given as

$$\int_{\zeta_1}^{\zeta_2} N(\zeta) d\zeta = \int_{E_1}^{E_2} N(E) dE = \int_{\zeta_1}^{\zeta_2} N(E) \frac{dE}{d\zeta} d\zeta \xrightarrow{\text{yields}} N(E) = N(\zeta) \left(\frac{dE}{d\zeta}\right)^{-1} \quad (22)$$

Supplementary Figure 10: **TOF spectra for energy calibration.** TOF spectra collected at different sample bias. The spectrum moves to higher channel numbers (lower flight times) with increasing bias as electrons are starting from the sample with additional energy equal to $(-e) \cdot (\text{sample bias})$. The secondary electron peak begins at a channel corresponding to the lowest energy electron $(-e \cdot (\text{sample bias}))$ which gives us the relation between channel number (TOF) and kinetic energy corresponding to the channel number.

Supplementary Figure 11: **Determination of the edge channel.** The TOF spectrum with sample bias of -20 V. The channel number that corresponds to the energy $(-e) \cdot (\text{sample bias}) = 20 \text{ eV}$ is found by fitting a line (red) to the falling edge of the secondary peak.

Supplementary Figure 12: **Energy conversion function.** Plot of the channel number versus (sample bias)^(-1/2) that is used to determine the energy conversion function. The data is fit to a fourth order polynomial from which the energy corresponding to each channel can be found.

The energy converted data is smoothened using a forward averaging algorithm which can be expressed as

$$N_{avg}(E) = \frac{1}{n} \sum_E^{E+\delta E} N(E) \quad (23)$$

where $N(E)$ is the number of counts in energy bin E , δE is the smoothing step size, n is the number of energy bins from E to $E + \delta E$ and $N_{avg}(E)$ is the new counts in the energy bin E . In the present paper, data has been smoothened with $\delta E = 0.25$ eV.

Supplementary Note 7: Calculation of the transport efficiency of TOF spectrometer at U.T. Arlington

In order to calculate the transport efficiency ($\eta(E)$) of the TOF spectrometer used in the experiment, the trajectory of the electrons from the sample to the MCP was simulated using SIMION® 8.1 ion and electron optics simulator [22]. The TOF spectrometer as shown in supplementary figure 8 was constructed in SIMION® 8.1 to accurately represent the dimensions, applied electric fields and applied magnetic fields. The sample was biased to -0.5 V with respect to the TOF tube. The $\mathbf{E} \times \mathbf{B}$ plates were biased to -2.79 V and +3.19 V which is equal to the values used in the present experiment. 2000 mono energetic electrons were flown from the sample and the number of electrons which reached the MCP was recorded as a function of electron energy. This is shown in supplementary figure 13. The data was fit to a function to deduce $\eta(E)$ and was used to correct the PAES spectra. It can be seen that there is an appreciable drop in efficiency only below 1 eV and above 500 eV. Between 1 eV and 500 eV the transport efficiency is close to 1. Thus, the correction applied to the PAES spectrum does not significantly alter the shape or intensity of Auger peaks analysed

Supplementary Figure 13: **Transport efficiency.** Transport efficiency of the TOF spectrometer used in the experiment was calculated using SIMION® 8.1 [22]. 2000 mono energetic electrons were flown from the sample biased at -0.5 V with respect to the ground and the number of electrons reaching the electron detector was recorded as a function of electron energy. The data was fit to a function to deduce $\eta(E)$ and the spectrum $N(E)$ was corrected to $N(E)/\eta(E)$.

Supplementary References

1. E. Boronski and R. M. Nieminen, Electron-positron density-functional theory. *Physical Review B* **34**, 3820-3831 (1986).
2. B. Chakraborty and R. W. Siegel, Electron and positron response to atomic defects in solids: A theoretical study of the monovacancy and divacancy in aluminium. *Physical Review B* **27**, 4535-4552 (1983).
3. P. E. Blochl, Projector augmented-wave method. *Physical Review B* **50**, 17953-17979 (1994)
4. G. Kresse, Efficient iterative schemes for ab initio total-energy calculations using a plane-wave basis set. *Physical Review B* **54**, 11169-11186 (1996).
5. G. Kresse and J. Furthmüller, Efficiency of ab-initio total energy calculations for metals and semiconductors using a plane-wave basis set. *Computational Materials Science* **6**, 15-50 (1996).
6. G. Kresse, D. Joubert, From ultrasoft pseudopotentials to the projector augmented-wave method. *Physical Review B* **59**, 1758-1775 (1999).
7. J. Klimeš, D. R. Bowler, and A. Michaelides, Chemical accuracy for the van derWaals density functional. *Journal of Physics: Condensed Matter* **22**, 022201 (2010)
8. J. Klimeš, D. R. Bowler, and A. Michaelides, Van der Waals density functionals applied to solids. *Physical Review B* **83**, 195131 (2011)
9. G. Graziano, J. Klimeš, F. Fernandez-Alonso, and A. Michaelides, Improved description of soft layered materials with van der Waals density functional theory. *Journal of Physics: Condensed Matter* **24**, 424216 (2012).
10. Y. Fukaya, S. Entani, S. Sakai, I. Mochizuki, K. Wada, T. Hyodo, and S.-i. Shamoto, Spacing between graphene and metal substrates studied with total-reflection high-energy positron diffraction. *Carbon* **103**, 1-4 (2016).
11. I. Makkonen, M. Hakala, and M. J. Puska, Modeling the momentum distributions of annihilating electron-positron pairs in solids. *Physical Review B* **73**, 035103 (2006).
12. A. Rubaszek, Electron-positron enhancement factors at a metal surface: Aluminum. *Physical Review B* **44**, 10857-10868 (1991).
13. A. Rubaszek, Z. Szotek and W. M. Temmerman, Nonlocal electron-positron correlations in solids within the weighted density approximation. *Physical Review B* **58**, 11285-11302 (1998).
14. H. Przybylski and G. Borstel, Nonlocal Density Approximation to Exchange and Correlation in Self-Consistent Bandstructure Calculations: Application to Cu. *Solid State Communications* **49**, 317-321 (1984).
15. H. Przybylski and G. Borstel, Nonlocal Density Approximation to Exchange and Correlation: Ground State Properties of Solid Copper and Vanadium. *Solid State Communications* **52**, 713-716 (1984).
16. N. D. Drummond, P. López Ríos, R. J. Needs and C. J. Pickard, Quantum Monte Carlo Study of a Positron in an Electron Gas. *Physical Review Letters* **107**, 207402 (2011).
17. H. D. Hagstrum, Theory of Auger Ejection of Electrons from Metals by Ions.

- Physical Review* **96**, 336-365 (1954).
18. H. D. Hagstrum, Theory of Auger Neutralization of Ions at the Surface of a Diamond-Type Semiconductor. *Physical Review* **122**, 83-113 (1961).
 19. J. J. Lander, Auger Peaks in the Energy Spectra of Secondary Electrons from Various Materials. *Physical Review* **91**, 1382-1387 (1953).
 20. R. A. Ferrell, Theory of Positron Annihilation in Solids. *Reviews of Modern Physics* **28**, 308-337 (1956).
 21. D. Manura & D. Dahl, SIMION (R) 8.0 User Manual (Scientific Instrument Services, Inc. Ringoes, NJ 08551, <<http://simion.com/>>, January 2008).
 22. M. P. Seah, Quantitative Auger electron spectroscopies and energy ranges *Surface Science* **32**, 703-728 (1972).
 23. S. Mukherjee et al. Time of flight spectrometer for background-free positron annihilation induced Auger electron spectroscopy *Review of Scientific Instruments* **87**, 035114 (2016).